# Electromagnetic compatibility issues in hybrid wired and wireless industrial networks

**Piotr Gaj** *, **Michał Maćkowski**

Department of Automatic Control, Electronics and Computer Science, Silesian University of Technology, Gliwice, Poland

* piotr.gaj@polsl.pl

## Abstract

Industrial networks are currently the only communication means designed for real-time systems used in industry. Networked control systems (NCS) are still important and commonly used type of such systems operating on shop floor. As a computerized node of NCS, a programmable logic controller (PLC) is usually used. In most cases, contemporary devices of such kind are equipped with more than one network interface of various types. Typically, only one interface is activated in NCS. Sometimes, the other is used for communication between NCS and supervisory systems. Occasionally, it is additionally involved in the data transmission in the factory IT systems. In general, however, using a single network interface is a more common solution. In this paper, the mutual utilization of more than one interface is discussed in order to back up the NCS network and to manage the node-related traffic within the scope of higher level services. The question of dependability of such a system from the electromagnetic compatibility point of view is discussed. The example is provided based on Profinet via wired and wireless connection.

## 1 Introduction

NCS is a well-known and commonly used architecture of computer-aided automation systems. The main components are 'intelligent' nodes and communication facilities. Such elements have to be capable to run the time-deterministic data processing and traffic. In order to provide the real-time (RT) data transmission, computer networks of industrial sort are used with their physical media and deterministic protocols. In typical solutions, such communication is accomplished by fieldbus or RT Ethernet (RTE) technologies. Modern approaches are focused on RTE or on next generations of industrial networking [1]. Despite using a particular technology, only one type of network is usually utilized. Even if network redundancy is exploited, the given network media with their node interfaces are run as redundant ones. Thus, only one type of media and communication interfaces are used. Potentially, contemporary nodes are able to manage more than one network interface, taking into consideration both media types and protocols. Programmable controllers are representative examples of NCS nodes [2]. According to the hardware and functional models from the IEC 61131 standard, such devices should be equipped with multiple communication interfaces. In practice, many of available devices comply with such guidance. Moreover, to allow automation

**Data Availability Statement:** All relevant data are within the manuscript and its Supporting Information files.

**Funding:** This work was supported by statutory funds for researchers (grant no. BK-213/RAU2/

2018) of the Institute of Informatics, Silesian University of Technology, Gliwice, Poland. The funders had no role in study design, data collection and analysis, decision to publish, or preparation of the manuscript.

**Competing interests:** The authors have declared that no competing interests exist.

networks to support different protocols simultaneously, the IEC working group produced the IEC 62439 standard. This standard defines redundancy methods applicable to most industrial networks. The methods differ in terms of topology and the recovery time. Moreover, there is another standard, IEC 61499, prepared for designing and programming the distributed control systems (DCS) at a higher level of abstraction, without considering specific network interfaces but with the possibility of using multiple ones in the system.

Therefore, various network interfaces are embedded in the controller CPU (MPU according to the terms of the above- mentioned standard) or can be attached as an extension of its local configuration. Both structures produce extended network functionalities which can potentially be utilized. Usually, such extra functions are turned off when they are available or are not even taken into consideration. In particular cases, such additional network interfaces are used to build up integration of the given node with other systems. Nowadays, the promoted way to assure a link between NCS and other systems is to use a unified communication layer based on the coherent network technology i.e., RTE. Therefore, in modern approaches, the development of inter-networking bridge-type functionalities in NCS nodes is rare, except when keeping the compatibility in heterogeneous environments. Hence, in practice there are many devices of NCS which are or can be equipped with additional network interfaces, and which are not used at all or are used only in one, quite simple integration service. In this paper, it is proposed to make such interfaces involved in NCS dataflow as additional communication means to achieve two goals:

1. network backup,

2. building a platform for convergent services.

To accomplish the first goal, the authors aimed at increasing the dependability of the industrial network and increasing a general safety level of the NCS. In similar cases, the redundancy technology is usually considered, which is technically justified. However, there is a situation when the classic redundancy could fail. The redundancy of the network media and its infrastructure **does not fully protect from radiated and conducted disturbances** existing very often in an industrial environment. Even if the wires are led in different routes, there is a possibility that electromagnetic disturbances can reach the whole redundant infrastructure or that the source of disturbances is located near the node. In case of problems with shielding (e.g., shield damage and abrasion, lack of proper grounding connection, detachment at plug, etc.) and exposing a network cable to an electromagnetic field (EMF) as well as exposing the cable to insulation scrapes and undesirable shortcuts or junctions, the network is prone to disturbances. Moreover, the wired transmission is prone to be disturbed even if the cable structure is good, as it is shown in the next part of the paper. The EMC (ElectroMagnetic Compatibility) issues are especially important when the used network is of RTE type. It is because the signal has a relatively high frequency on the physical media and may become susceptible to interference. In section 3, it is shown that using the RTE network via standard wiring as a main link together with wireless connection as a redundant link can provide a fault-tolerant solution when radiated and conducted disturbances occur.

The second aim of the paper is to provide the concept of supplementary utilization of the additional link. It is proposed that a wireless interface is used as a platform for dynamic management of internal system traffic and external integration services. In section 4, it is shown that based on the spare interfaces, one can build a mutual communication solution involved in data transmission in both control and factory IT systems. Thanks to that, not only the backup function is assured but also additional services can be run on all networks being involved in the traffic management. It is especially useful when mobile services are required in local area of NCS operation.

All in all, the authors present a technically-supported motivation for using all accessible interfaces of NCS controllers in order to increase system dependability as well as the possibility to add or modernize local services. The wireless network is considered to be a temporary backup in case of a failure, to sustain selected system functionalities in a safe mode. The presented results show that this approach is reasonable from the EMC point of view and can be used together with the other networking concepts, not instead of them. The novelty lies in considering the mixed networking approach as a way of increasing resistance to EMI (Electro-Magnetic Interferences). The main reason for using a wireless network is to ensure the totally different media and enable to build the spare channel easily, which is especially valid if the NCS exists already and is intended to be upgraded.

Usually, the usage of a wireless link is discouraged throughout NCS due to the RT issues. There is some other research available on this topic [3–8], including scheduling issues [9–11]. In the paper, the authors do not consider the RT capability of such a link, and do not want to use it instead of the cable network. Despite the RT flaws, the wireless networks are recently used in industrial informatics. There are some modern concepts such as Wireless Sensor and Actuator Networks (WSAN), Cyber Physical Systems (CPS), Industrial Internet of Things (IoT), and other ones which use the industrial networks of the so-called third generation [1], [12] or Industry 4.0 based networking. In such cases the industrial wireless networks are used to make the system more pervasive and heterogeneous. Thus, they can play a significant role in the modern systems designing. Nowadays, the consideration of the wireless communication is one of the main trend in this domain, including hybrid solutions of wired and wireless networking.

The solution proposed in the paper has some distinctive features. Although, the usage of mixed networking is not completely new, the application of mixed-media networking is rare in the context of increasing immunity to EMC disturbances. As will be presented, the proposed solution does not eliminate the problem of EMI, but can improve the resistance of communication to electromagnetic disturbances. In industrial environments, the problem of interferences is very common and has a random and hardly repetitive character. By using the laboratory equipment it was possible to reconstruct the conditions of industrial factory and perform the tests. The conducted tests were long term and were carried out for different conditions (varying level and duration of the interference, different frequency range etc.)

## 2 Related work

Using two or more varied networks in order to assure the communication stability is not a new concept. Such approach is commonly known and used in practice. However, the approach proposed by the authors is a new one. The difference refers to the context of usage. Typically, the surplus networks act as a backup, which means they are designated for either getting access to the external zone e.g., the Internet, or sustaining the connection within the public network environment. The main goal in such a case is either diversification of service providers or multiplication of the available paths through the external networks. The spare network is activated when the main network is damaged or requested services are unavailable on it. As a sample, the wired and GPRS connection through the Internet presented in [13] can be used. In such a case both networks work together. One is the main network, and the other is a spare. Thus, when the given network provider fails, the other one can keep the connection (column A in Table 1).

The other utilization of the mentioned structure is the integration of heterogeneous elements and subsystems. The multi-network and hybrid solutions can make the system more flexible, resilient and pervasive, and allow creating communicatively coherent but

**Table 1. The comparison of available mixed networks concepts in relation to the proposed method.**

| | | Available concepts and schemas | | | | |
|---|---|---|---|---|---|---|
| | | A | B | C | D | E |
| | | [13] | [14] | [15,16] | [22–25] | **proposed one** |
| **Advantages** | **dependability** | | | | improved | improved |
| | **efficiency** | | | increased | | |
| | **spare access** | available | | | | available |
| | **integration** | possible | possible | | possible | |
| | **EMI resistance** | | | | | increased |

heterogeneous NCS. The sample of research on utilization of appliances with various network interfaces is presented in [14] (column B in Table 1).

In the literature, there are some attempts to utilize the redundant network architecture to increase the efficiency of the dataflow (column C in Table 1). The solution proposed in [15] is focused on utilization of the spare bus, and the main goal is to improve an overall throughput of the redundant channel in case of normal operation. In such an approach, the typical network redundancy is discussed, so the same kind of network is used. The dependability aspects are not considered at all, and the research is focused on the alternative utilization of such architecture. Moreover, the immunity against EMF interferences is similar to the non-redundant solution, if disruption affects both networks and a node.

The issues of radiated and conducted disturbances are considered in [16]. The concept of dynamic balancing of the traffic in two independent Modbus interfaces is proposed in this work. Hence, the utilization of the same kind of network is taken into account. It allows using the spare interface to build a redundant solution. Nonetheless, in the presented methodology, the link is prone to be corrupted by EMI, as previously.

As for wireless only networking, to avoid common problems of communication in industrial environments, a various methods of network self-adaptation are used as frequency hopping, adaptive routing, multichannel transmission, or dynamic channel allocation (e.g., RPL, TSCH) [17,18] as well as some based on them standardized protocols as IEEE802.11, ZigBee, WirelessHART and the ISA100.11a [19], IEEE802.15.4 [20]. The interesting solution of the multichannel protocol and communication is proposed in [21]. Authors using channel hopping and channel adaptation mutually to improve the network performance and link quality. All of these solutions are not mixed with cable networking, so the characteristics of EMI resistance is limited to the wireless transmission, without option to overcome its susceptibilities.

In this paper, the authors propose using at least two network interfaces with the completely different media: wired and wireless (column E in Table 1). Similar attempts of networks integration are presented as a hybrid network in the FlexWare [22] and VAN [23] concepts. In both cases, the main goal of coexistence and collaboration of such networks is mobility, reduction of wiring, increase in flexibility, and integration within the domain of wide area networks (column D in Table 1). The same approach is presented in [24]. The inclusion of the Profinet into WISA concept is presented in [25]. However, the mutual collaboration with dynamic traffic management in the research context of EMC issues is not commonly presented and known.

The Table 1 contains some selected advantages of the schemes and concepts existing in the literature vs. the proposed concept, based on which our motivations are highlighted.

The quantitative background of this qualitative sheet is based on the analysis and conclusions presented in the cited research. The features such as dependability, spare access, and integration possibilities are not strictly measurable. In this case they were estimated by the potential ability of the given solution to make a positive impact on the given system feature.

Network efficiency can be measured in various ways (throughput, time/band utilization, latency, etc.). Researchers usually consider throughput as a measure, and in this case the throughput increase was confirmed. EMI resistance has not been considered in the cited studies, but this is the purpose of our work, and the obtained results confirm its increase.

To sum up, the current work is focused on some protocols dependent features of mixed network architectures. The main point of the presented approach is to show that the proposed type of networking can increase the safety level from the EMC point of view. The NCS communication with mixed wire and wireless networking is a reasonable solution and can secure the internal Profinet connection. The same is true for other RTE protocols by implication. Usually, in order to increase the safety level, the redundant approach based on the same network technology is used. There are no solutions where different media are used to build the redundant channel, and even if some exist, there is no proof that the usage of the different media can increase the dependability level in practice. Thus, this is the reason why the presented approach is new and could be useful for NCS designers.

## 3 Communication distractions

There are many factors which can impact the proper operational state of the computer networks in the industrial environment. Network media are prone to be exposed to electromagnetic field as well as to be physically broken and influenced by either undesired junctions or lack of requested ones.

The contemporary environments in which electronic devices operate are quite cluttered from a connectivity point of view. There are many residential and industrial appliances which generate electronic noise. The reliable research on this issue can be found in [26]. Each type of area is disturbed, and particularly the industrial ones. According to [26], for industrial area the typical frequency of radio (RF) noise is between 80 MHz and 1 GHz. Above this frequency the noise power drops due to lower power of sources and higher absorption. Such disruptions of higher RF are especially important when using wireless solutions.

### 3.1 Threats

There are two typical EMC threats which can affect the computer network. The first one is EMF of strong enough intensity. The second one is a parasitic injection of undesirable signal into the shield.

The communication technologies are theoretically secured up to the normalized levels of the above-mentioned disturbances, but not in practice. In a launched NCS, some small defects can arrive due to usage issues as well as wear and tear. The typical defects are shield abrasion at bending points and wrong connection at the plug site. They appear due to vibrations, mechanical factors or other environmental impacts.

Furthermore, there is an additional factor which can impact the real behavior of considered transmission. It is the spatial range and place of the disturbance injection. The location of the disturbance source in relation to the distance from the communication node and the width of the disturbance effect can make a big difference to the system performance.

### 3.2 Research background

There are some standards which deal with EMC issues and specify the levels of the mentioned disturbances. Moreover, these standards are harmonized with the EU EMC Directive 2004/108/EC with which the products must comply as a requirement to be introduced to the EU market.

The programmable logic controllers and the communication protocol that were used during the tests are generally used in an industrial environment. Therefore, the research was performed in accordance with the requirements of the industrial environment standard IEC 61000-6-2:2005 "Generic standards–Immunity for industrial environments". Generic standards specify environment conditions, and set minimal EMI resistance that equipment in this environment must meet. Moreover, these standards also provide the general and fundamental rules for meeting the EMC requirements. The basic standard includes the test set up, test procedure, acceptable error level, and accuracy of measurements. The IEC 61000-4-x series of standards are the best known examples for basic standards.

The authors are aware that in real-life and normal operation, equipment is subjected to a number of electromagnetic disturbances occurring at the same time, for example: radiated fields from two or more devices simultaneously transmit a continuous radiated field, a fast transient burst (EFT) or electrostatic discharge (ESD). Simultaneous radio-frequency disturbances can cause more unusual and unexpected interference problems by intermodulating within electronic devices. Moreover, real world sources of RF interference have a huge possible range of modulation frequencies, but normally during immunity tests (according to 61000-4-3 and 61000-4-6) only a AM 1 kHz modulation frequency is used [27]. In laboratory conditions, however, it is difficult to reproduce such real-life conditions while maintaining repeatability of testing. The tests, where the type and range of modulation frequencies are chosen to cover the range over which the equipment concerned is susceptible are mostly used by certain military organizations, but are not yet commonplace during industrial tests.

The authors of the research have made efforts to properly adjust the test conditions when testing the immunity of communication interfaces to EMC disturbances. Chapters 4.3.4 and 4.3.6 discuss how the parameters such as frequency step size and dwell time (relevance indicator measuring the time the user remains at a search result after a start the disturbances) were selected.

Additionally, the wiring guidance from the Profinet standard exists. It refers to the cable type, its length, shield, connectors, grounding and equipotential bonding [28]. The authors assumed that these regulations fully define the properly constructed wiring of the Profinet network.

The signaled issue of disturbance influence in various spatial range and injection location is not explicitly stated in the documents mentioned above.

## 3.3 Research on EMI

To reduce the influences of EMI, the network composed of two different physical channels was proposed during the research. Thus, the main aim of the research was to test the effects of electromagnetic disturbances on the transmission of data between two PLC controllers. Two cases were considered: in the first case, the transmission between two devices took place via wire and in the second case, the transmission was via a radio Wi-Fi network.

The testing procedures and conditions are precisely defined in the above-mentioned EMC and Profinet standards as well, and they have been used in the presented research. Conditions which go beyond the values and structures derived from these norms and directives are not considered in this paper.

**3.3.1 NCS structure.** The Simatic S7-300 controllers were used as system nodes with a Profinet IO interface. The software was especially designed for the testing purpose. The communication failures were detected on the real-time application level in reference to the disturbance frequency. The idea of examining the connection was based on an algorithm in which a request and response are being sent cyclically between a client and server node. During such a

transaction, the content of the data structure was transferred from one node to another. The standard diagnostic of Profinet communication blocks was not used due to the relatively long time of error detection. The authors were not interested in detecting the communication which had been totally lost, but rather in detecting preliminary signs of communication problems caused by forced disruptions. In the paper, the main issue are not the temporal characteristics of dataflow and device functions but rather the connections reliability between controllers, from the RT application point of view. The deterministic protocols, as Profinet, operate in the cyclic way, where network actions are synchronized and temporal characteristics of the traffic are quasi-constant (with low jitter). Any disturbance of this cycle can violate the RT constrains and can make a negative impact to the application. Therefore, the most important measure of the connection reliability is a discrete status: operational/failed. The temporal characteristics of errors occurring at the given frequency have not been registered, because it is useless from the RT application point of view. Independently from the types of errors distribution in time, it always means non-operational state of the channel. Such characteristics could be useful where non-RT networks and non-RT applications are used.

The authors point out the possibilities for improving the channel dependability. Therefore, the communication disturbance was detected by measuring the response time. It was assumed that if the reply comes with a delay of 100% longer than the average response time, then communication interferences occurred. After detection, the request was cancelled and started again. Requests were sent every 1500 ms, where 1000 ms was a gap between requests and 300–500 ms was a timeout, depending on the conditions of the given test. The average response time on the not disturbed media was 50 ms. Thus, the error was registered when the response time was at least 100 ms. Detected events were stored in the memory as records with an event number and its timestamp with a resolution of 10 ms.

The connection both via cable and via IEEE 802.11b/g (WiFi access points) was used. The standard shielded cable (AWG 22) designed for industrial Ethernet was used. It is a cable suitable for the Profinet network. The tested segment had approx. 30 m. As plugs, the special connectors designed for Profinet were used. They had a metalized coat and a dedicated terminal for a Profinet cable. The wire segment was prepared with a professional stripper. The Profinet IO settings were fixed to a typical cycle of 1ms.

The wireless connection was run with a Moxa WiFi Access Point (AP), special outdoor antennas with 14 dB gain, with a short distance. The used frequency was 2.4 GHz without channel hopping and without other mechanisms of spectrum scattering. Only the media was tested, not the nodes and their interfaces. It was assumed that in practice the nodes are closed in a shielded zone, e.g., in a control cabinet.

The general schema of the tested structure of NCS is presented in Fig 1.

**3.3.2 Test-bench.** In order to generate EM disturbances, the following EMC measurement equipment was used:

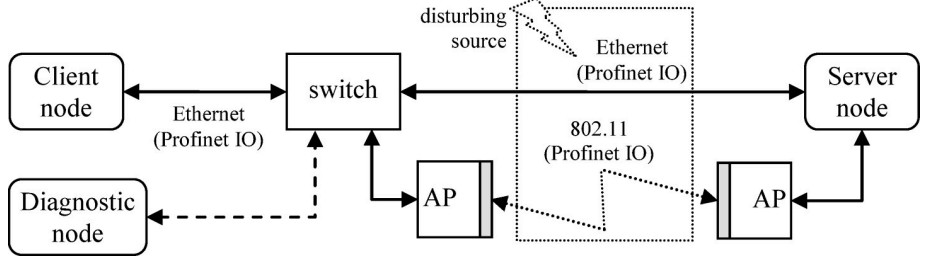

**Fig 1. The general schema of the test-bench.**

a. the EM clamp to make conducted disturbances induced inside the cable,

b. the Coupling Decoupling Network (CDN RJ45) to generate conducted disturbances in the shield,

c. the GTEM (Gigahertz Transverse Electro Magnetic) cell to induce interferences inside the specified area.

The clamp and CDN are able to generate disturbances in the range of 150 kHz to 230 MHz. The GTEM cell can generate a stable EMF (in general, homogeneous and uniform) from 80 MHz to 4 GHz. Both ranges are presented in Fig 2 with a nonlinear scale.

For the lower frequency range from 150 kHz to 230 MHz, the test was conducted according to the basic standard IEC 61000-4-6. Such standard relates to the conducted immunity requirements of electrical and electronic equipment to electromagnetic disturbances coming from intended RF transmitters in the frequency range 150 kHz up to 80 MHz. In specific cases, this standard also allows to extend the measurement range to 230 MHz. For the purpose of research, it was decided to carry out tests for the wider range. Other test parameters were also defined in accordance with the standard. The IEC 61000–4 standards impose the use of amplitude modulated (AM) signals (using frequency modulated (FM) signals does not generally produce any additional susceptibilities except in special cases). For AM, a 1 kHz sinewave is normally used, with some product-specific exceptions. These standards refer to the specified level of unmodulated signal, which is then modulated at 80% depth and the disturbances level is set to 10 V (industrial environment requirements).

During the research, the influence of conducted disturbances on the transmission wire was examined. The schema of the test bench is shown in Fig 3. One of the elements of the test bench was an EM-clamp (and alternatively the CDN) consisting of a tube of split ferrite rings of two different grades, which can be clamped over the cable under test in a non-invasive way. The EM-clamp provides both inductive and capacitive coupling and can be used in the range 150 kHz to 230 MHz. During the test, the Rohde&Schwarz equipment for EMC such as signal generator, power amplifier, EM clamp were used.

In the second part of the research, the focus was on the impact of radiated disturbances (electromagnetic field) on wireless transmission between the two devices. The test was carried out per the guidelines specified in the standard test for radiated immunity–IEC 61000-4-3 "Radiated, radio-frequency, electromagnetic field immunity test". The test requires that the EUT (Equipment Under Test) operates satisfactorily when subjected to a strong radiated electromagnetic field (such might be created by cell phones and other intentional radiators, and RF noise that might be caused, inadvertently or otherwise, by industrial processes).

The test required a radiated RF field generated by an antenna in a shielded environment. The field was a pre-calibrated field, swept from 80 MHz to 4 GHz and with a dwell time

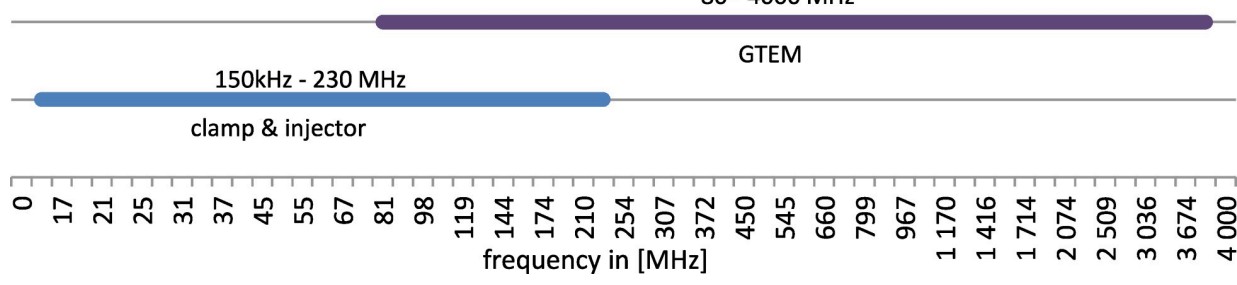

**Fig 2. The ranges of distracting EMF frequency.** Two devices were used.

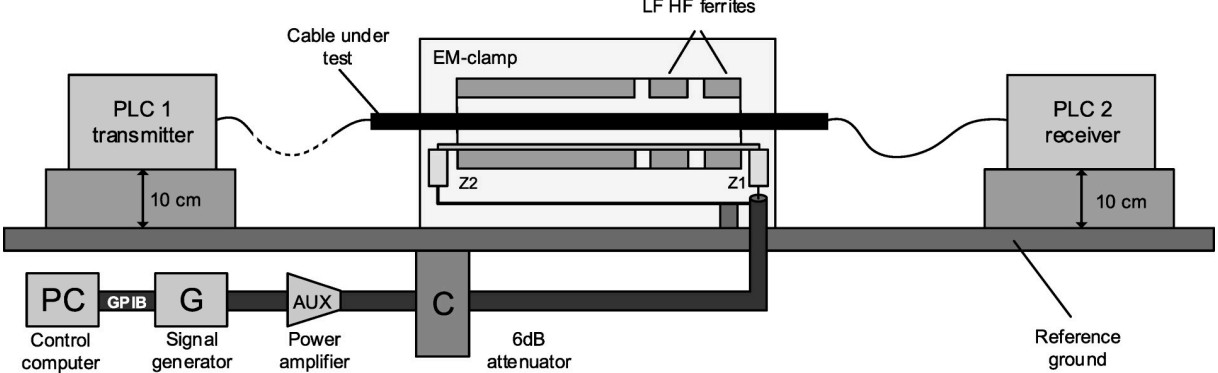

**Fig 3. The scheme of the first test bench.** It was used during the research in accordance with the standard 61000-4-6.

sufficient to allow the EUT to respond. The disturbance level was set for industrial environment (10 V/m).

During the research a GTEM cell was used, which is a test site for efficiently performing both radiated immunity and emissions testing in a single, controllable and shielded environment. Electromagnetic field is generated by the connection of an RF generator and power amplifier to the input point of the GTEM cell (Fig 4). The sets of the requirements for performing EMC testing in waveguide (GTEM and TEM cell) are also published in IEC 61000-4-20 "Testing and

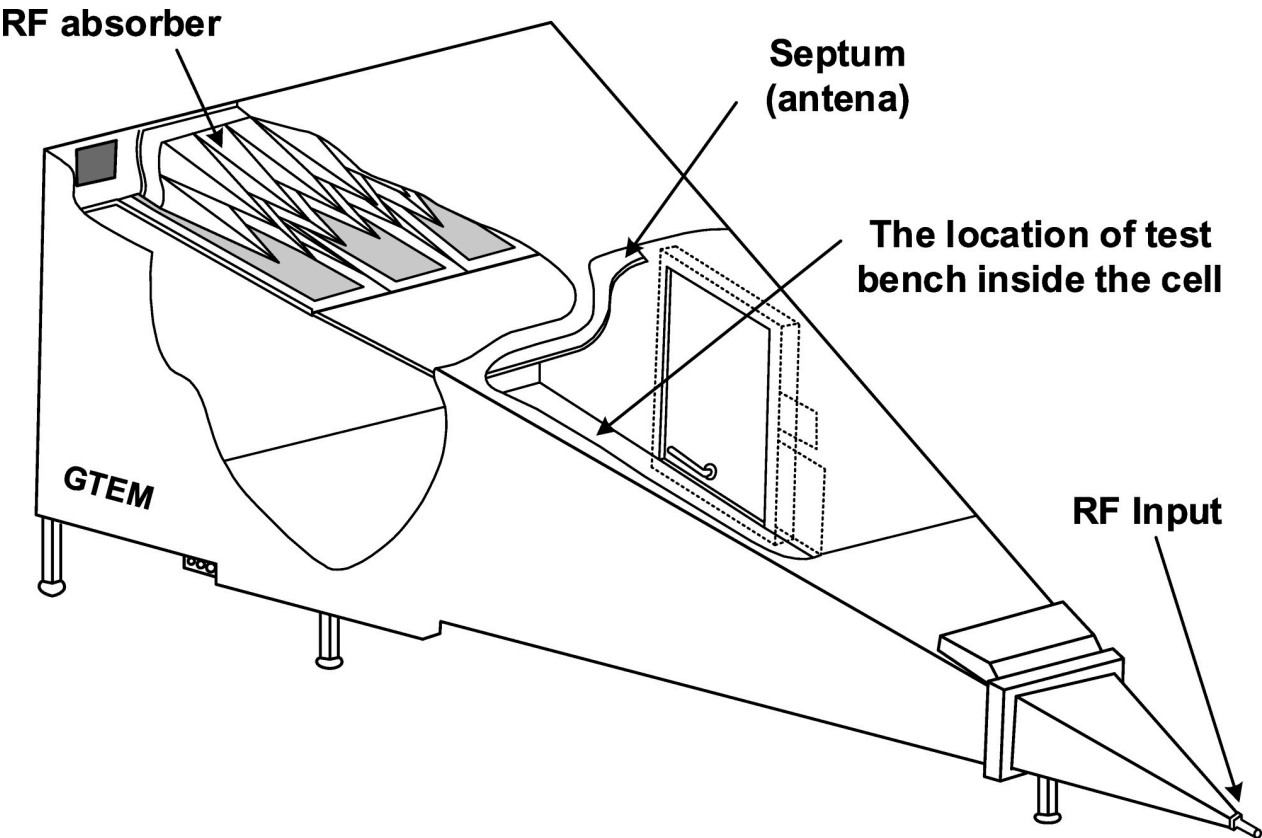

**Fig 4. The scheme of the second test bench.** It was used during the research in accordance with the standard 61000-4-3.

measurement techniques–Emission and immunity testing in transverse electromagnetic (TEM) waveguides". The requirements of this standard were also taken into consideration.

During the research, only two Wi-Fi antennas for communications were placed inside the GTEM cell. The rest of the hardware was placed outside the cell to minimize the impact of the electromagnetic field on the device.

**3.3.3 Test cases.**   For such a NCS structure and test-bench, several cases were selected to be tested with a wired network. All cases were tested with a proper shield, and additionally with a broken one. The issue with the broken shield was that there was no proper grounding connection and the equipotential bonding was interrupted. An unshielded cable was not tested as it is a wrong approach. The selected cases were as follows:

a.  A cable without damages. It was the normal operating state of the wired network.

b.  A cable with conducted disturbances injected into the shield. It was done by using CDN on the cable.

c.  A cable with conducted disturbances induced inside the cable. It was done by putting the cable inside the EM clamp.

d.  A cable within the area of EMF influence. The cable was placed inside the GTEM cell. Nodes were outside. Two cases were considered: with approx. 10% and approx. 90% of cable length under distractive (electromagnetic) field.

Additionally, two cases were selected for the wireless connection. Due to the character of media it was impossible to make conducted tests.

e.  Wireless connection without disruption. The test was performed in the area with EM noise of office kind.

f.  Wireless connection within the area of EMF influence. The test was run in the GTEM chamber. Only antennas were placed inside the chamber. The transmission was through the area of stable EMF with the constant intensity of 10 V/m. All Profinet settings were kept without changes.

**3.3.4 The cable connection.**   During the tests the dwell time of the interference was considered, because the signal must be applied for a sufficiently long time for the EUT to respond, thus it should be as long as the longest appropriate EUT time constant. The standard defines that the maximum step size allowed is 1% of the actual frequency and dwell time 2.88 seconds. Therefore, in order to carry out more detailed research, it was decided to reduce the frequency step and extend the dwell time so that the effect of disturbances on the quality of transmitted information could be observed.

It was assumed that the frequency step was linear and set to 500 kHz for the clamp and 10 MHz for the GTEM with a dwell time of 12 sec in both cases. Such setting produces a single measurement collection time of about 90 minutes. These parameters vary from the ones described in the 61000-4-3 and 61000-4-6 standards, where for example the frequency step size is equal to 1% of the previous frequency value. The smaller frequency step (especially for the higher frequency) allows increasing the measurement accuracy and extending the number of measuring data. Thus, for case (a) the measuring time was 90 min. No disturbances were registered in this case. For test (b), the measurement series revealed some weak points of the cable channel. It is presented in Fig 5.

During the disturbances in frequency above 140 MHz, the number of delayed transactions rose. However, there was no communication loss detected by Profinet low level diagnostics.

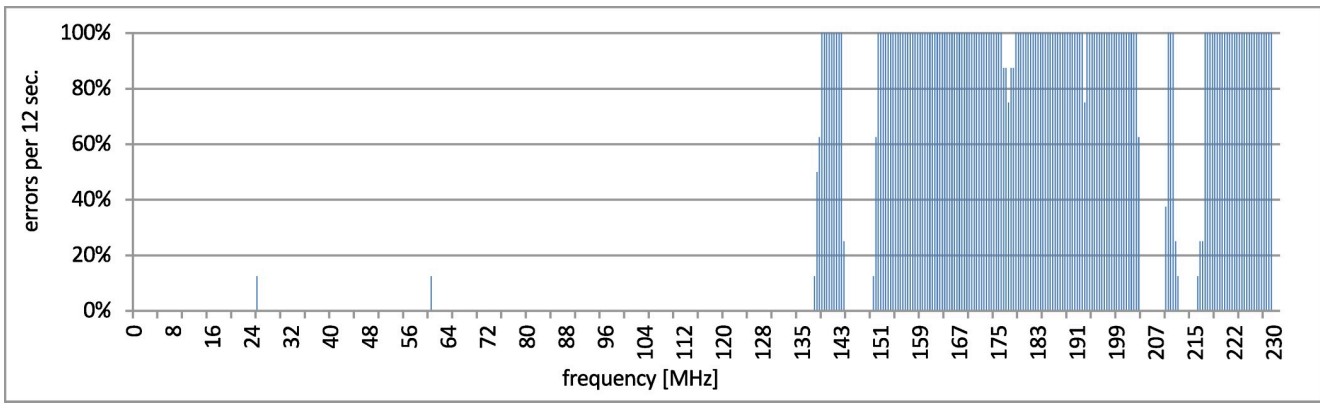

**Fig 5. Disturbances in case (b)–whole range.**

The observed cutoff of the number of errors at 8 (100% errors) comes from the dwell time. In the fixed 12 seconds window and with the assumed timeout (1.5 s), the maximum number of failed transactions is 8. The responsive range of above 140 MHz was tested with increased dwell time (20 s) of the frequency steps and the same timeout. It is presented in Fig 6. The cutoff is on approx. 13 (100% errors during dwell time).

The important thing is the fact of disturbance responsiveness and its scope of frequency.

In case (c), the measurements delivered a similar disturbance characteristic, however, it was narrower and had a slightly different shape. The susceptible range was 170.4–200.2 MHz and it was tested with reduced step to 250 kHz and increased dwell time to 20 s. The results are presented in Fig 7.

All the above tests were conducted with the proper cable. Tests of a damaged cable delivered slightly worse results. The similar tests of a wired connection were performed in the GTEM cell in the full range of 80 MHz up to 4 GHz–case (d). No negative induced influences had been found, even if the shield was connected in one point only and there was no equipotential line.

As can be seen in Figs 5–7, despite using the shielded cables during the tests, the transmission was still disrupted by electromagnetic disturbances. For the disturbances in frequency range from 140 MHz to 230 MHz, the physical layers were most prone to the EMC interferences. Similar results and theoretical background that support the presented approach were

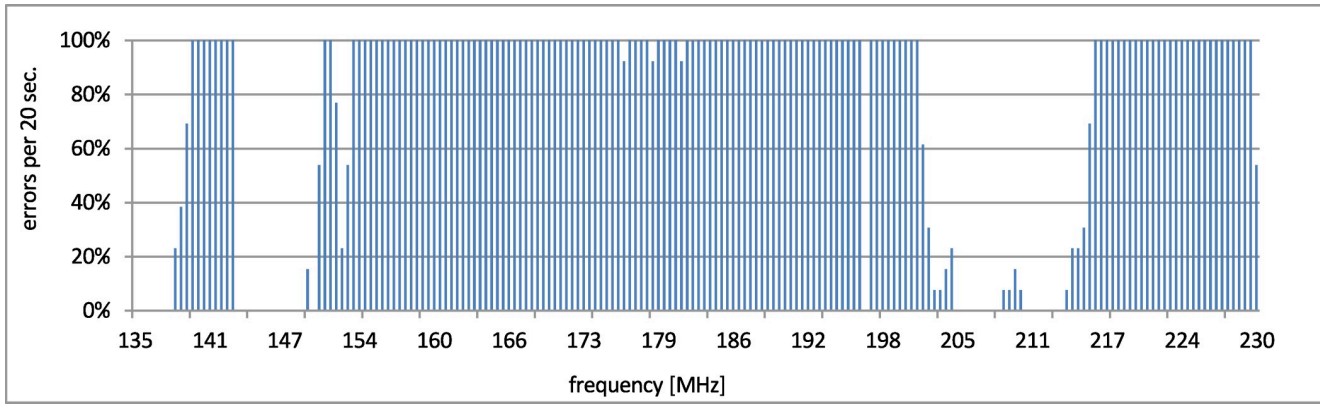

**Fig 6. Disturbances in case (b)–susceptible range.**

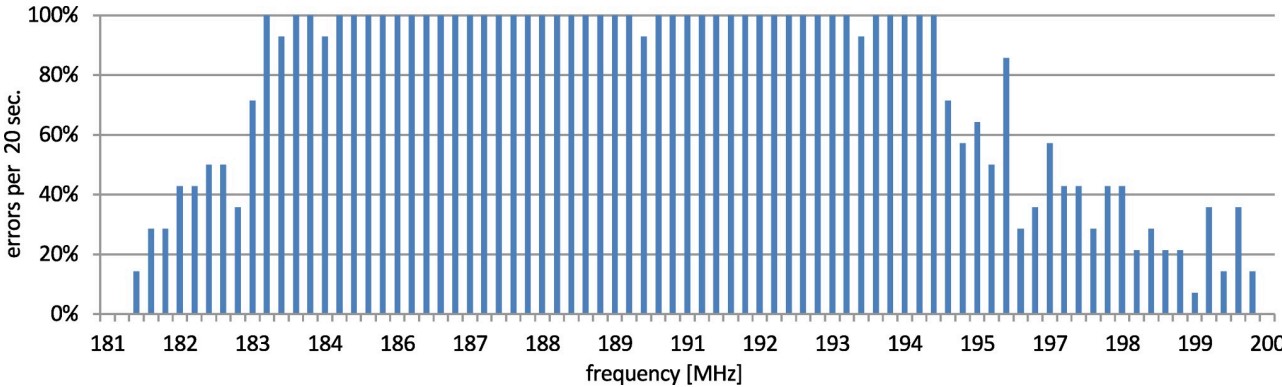

**Fig 7. Disturbances in case (c)–susceptible range.**

presented in papers [29,30]. There were numerous tests conducted and the interferences of data transmission occurred always for the same frequency of the disturbing signal (this process was repetitive).

**3.3.5 The location of the disturbance.** After identifying the susceptible ranges for case (b), the network was tested with the disturbance source located at a few location options. The simplified schema of such locations is presented in Fig 8. The short distance in this case is approx. 1 m and the long one is approx. 35 m. The plus and minus marks identify the disturbance susceptibility within this range. The EUT connector is the equipment under test side of CDN.

The conclusion from the above tests is that the connection is even more prone to disturbances if the source of disturbances is near the node. Such phenomenon is in accordance with the theory and it is well known [31,32]. It is included here only to note that the source location can have significant impact on the regular redundancy.

**3.3.6 The wireless connection.** The final thought from the cable testing is that it is quite easy to disturb a wired communication above 140 MHz. The expected results from testing the wireless channel are that it is vulnerable to EMI but not in the same range. At the beginning of the test, the communication via 802.11 was performed in a few series of 90 minutes without any disturbances. In this case (e), a few errors occurred, generally in random moments. It is assumed that the errors appeared due to the typical EM interference with the office-type

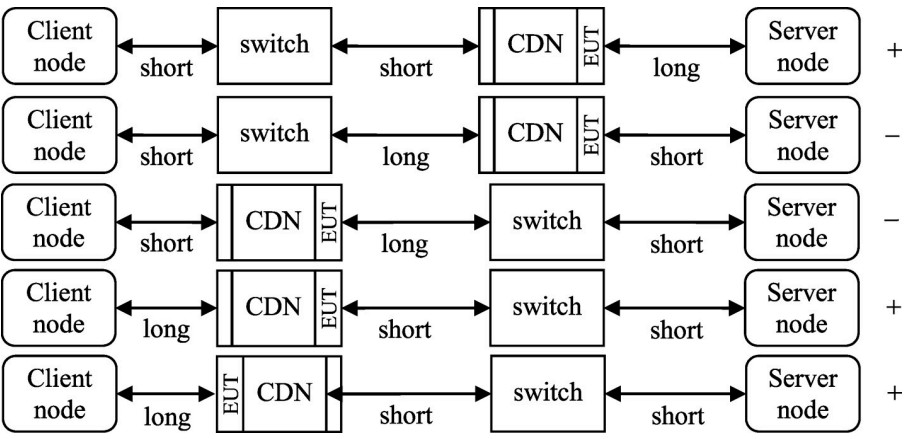

**Fig 8. Location of disruption source in relation to the node.**

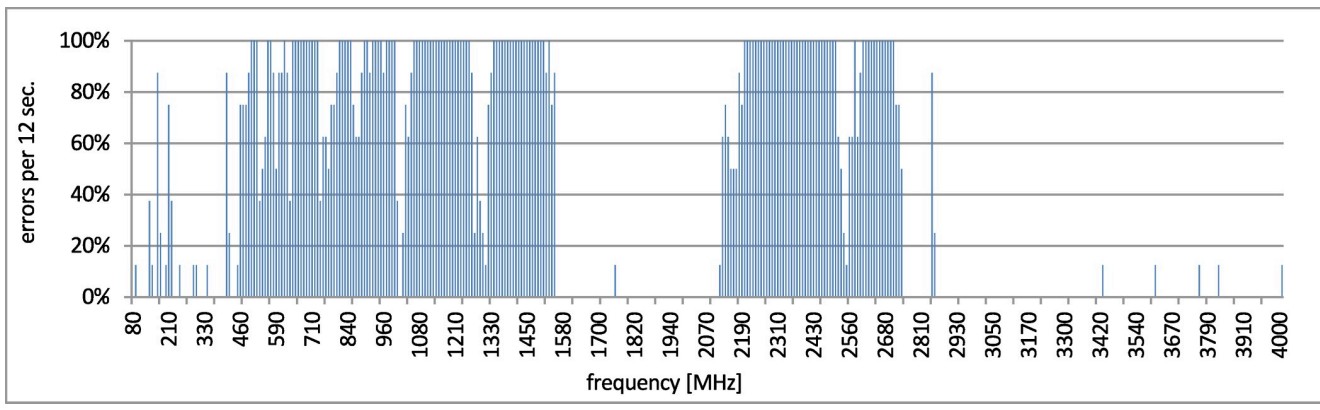

**Fig 9. Disturbances of wireless connection (f).**

environment. Next, the disturbances in the wireless connection (f) were checked from 80 MHz up to 4 GHz with the step of 1 MHz and a dwell time of 12 s. The result is presented in Fig 9.

The disturbance area starts approx. from 400 MHz up to 1.55 GHz and next from 2.1 GHz up to 2.75 GHz. The number of errors within the other ranges is low or none. Next, the interesting area of above 140 MHz up to 400 MHz was tested with the step of 1 MHz and a dwell time of 12 s. The results are presented in Fig 10.

The test was conducted in several series in order to confirm the reproducibility of results. Collected outcomes confirm that there are some narrow bands of frequencies in which the communication is disturbed. Disruptions in the given channel are repetitive. They vary somewhat along the changes of the used channel. This is presented in Fig 11. In order to reduce the influence of disruptions, some mechanisms of channel hopping would be useful.

**3.3.7 Observations.**   The communication failures observed in both media result from the interference phenomena between the physical signal used with the given media and the disturbance signal. As a result, in some ranges of the tested spectrum the areas with or without errors can be observed.

In case of cable and Wi-Fi usage, the different types of disturbance sources might exist. Thus, the type of stimulus is not the same. It is very likely that if one type of media is influenced by EMI, the second one is resistant at the same time to the same disturbance. It comes from the results presented above and it is presented in Fig 12 as a pictorial view. The dotted lines depict the communication failure range and the solid lines depict the failure-free or the occasional failure range.

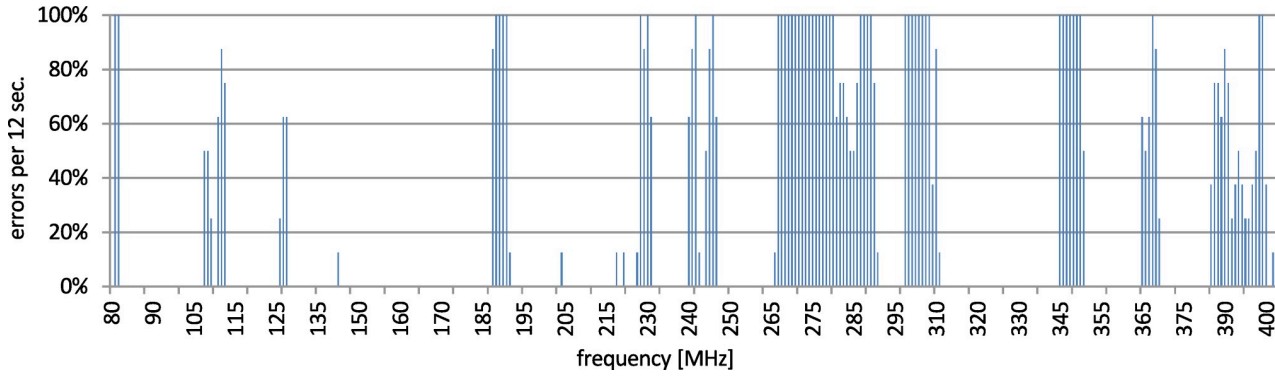

**Fig 10. Disturbances of 802.11 channel 1.**

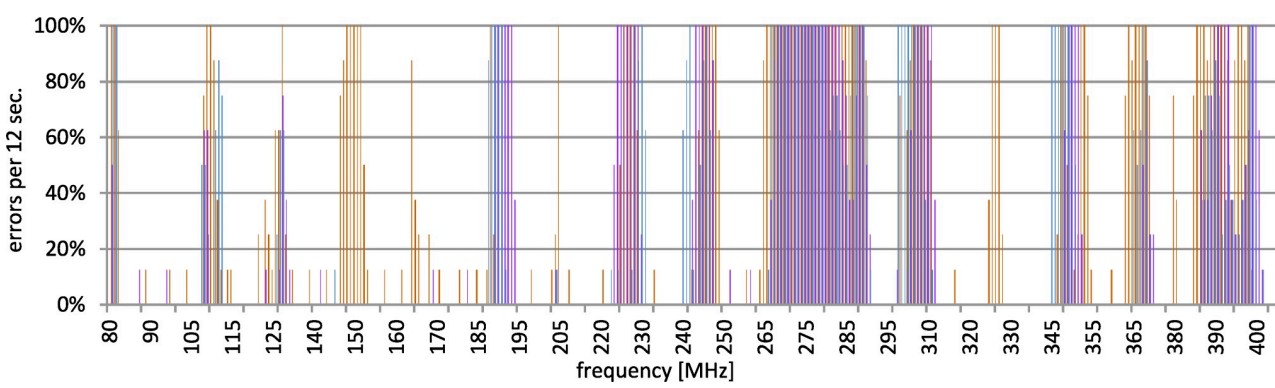

**Fig 11. Disturbances of 802.11.** Data presented for selected channels 1, 6, and 13.

Therefore, by using an industrial network with media of different types, one can avoid the common influence of the given disturbance source. The influence depends on the media type and the frequency as well as the type of signal propagation. Thus, different media assure different response to the same disruptions or eliminate some stimulus due to their nature.

It could be assumed that for the purpose of securing the range of 1–3 GHz the wireless media can be used. In practice, as it can be observed in Figs 9, 10 and 11, the complementary usage of both media is a good idea due to the different characteristic of resistance, but not due to mutual exclusiveness to disturbances. In the range 80 up to 400 MHz the interference on the both media is observed. Nonetheless, the interferences of the wireless network in this area are visible only in narrow bands and their intensity is lower than the intensity of disturbances in the wired network. E.g., in the specific range of 130–230 MHz the number of errors in the wireless channel was more than 6 times lower than in the wired one, as presented in the Table 2. All presented errors refer to the network transactions executed on the application layer and are counted according to the previously presented test assumptions. The layers of the protocol stack have not been analyzed. The Profinet protocol is a well-known and it is well-tested. The analysis of the internal behavior of the Profinet stack, and delivering e.g., BER (bit error rate) on the given layer, could characterize only the given instrumentation and implementation of the Profinet and can either confirm or deny its accordance to the well-known standard. In this research the cross-standard behavior is the aim, when the environment characteristics go beyond the industrial expectations formulated in each of the constituent network standards. The presented error rate is the summarized error rate for the whole frequency range, common for to both wired and wireless connection. This illustrates the overall failure difference in this range.

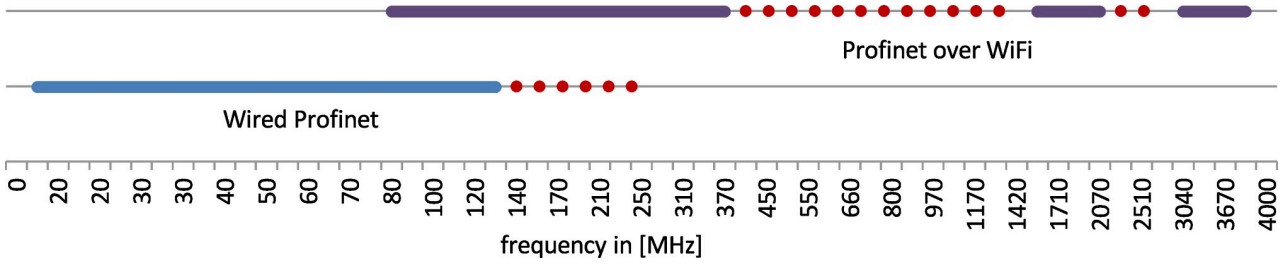

**Fig 12. The summary of the disturbed areas in a Profinet connection.**

**Table 2. The number of errors in the wired and wireless channel.**

| Common range 130–230 MHz | | | | | |
|---|---|---|---|---|---|
| Total number of transactions | | Total number of failed transactions | | PER (packet error rate) | |
| Wired | Wireless | Wired | Wireless | Wired | Wireless |
| 1600 | 800 | 1195 | 87 | 75% | 11% |
| | Variance | 11.54 | 4.81 | | |
| | Standard deviation | 3.40 | 2.19 | | |

Hence, it is expected that in this common range the wireless channels provides a more error-resistant solution with lower probability of error occurrence. However, the wireless channel cannot make the hybrid network fail-proof, but it can reduce the likelihood of failure. The wireless failures observed in this range could be additionally reduced by a channel management mechanism, e.g., as described in [33].

Generally, the Wi-Fi usage is limited by the given RT requirements of data transmission and open for establishing additional services. The limitations are the same as in Wireless Sensor Networks, Cyber Physical Systems, Internet of Things, and other concepts which use the third generations of industrial networks [1], [12]. Thus, the wireless connection can be considered as a backup channel, which is appropriate for creating a cheap network redundancy with the ability to keep the system alive when the main network is broken, and with potential facilities for increasing system convergence.

## 4 Concept of a multi-network logic channel

Involving an additional network solely for the purpose of building a second logical channel of NCS communication might not be a convincing solution for extending the system infrastructure. Thus, the authors signal a further benefit of the presented approach in this section of the paper. The utilization of an additional network for not EMC oriented purposes is widely considered in related publications, so here it is only mentioned as a concept of additional existing features.

The extra utilization of the wireless network can equip NCS with some new services. It is quite clear that the wireless network has a different transmission characteristic then a cable one. The security and safety issues as well as the real-time characteristics are not the same. It comes from the nature of signal propagation. Therefore, the wireless network cannot be simply used instead of an industrial network based on a cable. That is not the claim made in this paper.

In fact, different media means different networks based on different network interfaces. Such networks and their interfaces can be combined into one logical communication channel, from the application point of view. Hence, it is proposed to use available Ethernet ports to construct a logical channel running different and independent interfaces, and to follow the physically diverted networks for the purpose of communication needs of the node application.

### 4.1 Improving safety

The wireless network can be used as a spare channel for temporarily usage when the main cable is disrupted or down. The most important services (e.g., cyclic update) can be transferred to the wireless channel. Thanks to that, the system can be fault-proof in the scope of selected services. The management of such redirection can be based on the well-known mechanisms from a redundant network solution together with additional possibility of selecting the requested range of services.

It is assumed that in the proper conditions, the backup network is not used for the servicing of RT traffic. It is activated only when the disturbance level on the main network is greater than the assumed one. The detection of improper conditions can be executed by delay measurements, by counting errors within the time unit, or by the embedded diagnostic functions.

## 4.2 Extended features

When backup is not necessary, the proposed structure allows including some extended features to the system communication abilities, which can improve systems interconnectivity, inter-systems convergence, and utilization of additional interface and its free bandwidth.

It is not important to propose the given service. It depends on the local requirements of NCS. Potentially, the utilization scope could be divided into two groups:

1. Interconnectivity–data monitoring services (e.g., SCADA, MES, ERP), diagnostics, statistic measurements, database services, parametrization functions, etc. The good and innovative example in this group could be local and mobile HMI services. The communication scenario of such services can be implemented as a fixed schema, with activation/deactivation feature triggered by a disturbance detection unit.

2. Traffic balancing–the traffic in industrial networks can be divided into several classes. Usually, it is an RT, non-RT and free traffic classes. Thanks to the additional channel, the application can redirect a given type of traffic (e.g., non-RT and free traffic) from the main network to the second one. Thereby, the band allocation of the main channel could be lightened. Implementation of such feature can be done by extending a node application by a switch unit responsible for redirecting data to the given channel.

## 5 Conclusions

Based on the research results, it can be observed that the industrial RTE network composed of two different media and with the same protocol has better disturbance resistance then the same network with the given media separately. The usage of mixed networking is not new, however, the utilization of mixed-media networking is rare, especially in the context of EMC issues. The proposed structure does not eliminate the problem of EMI, but can improve EMI resistance of communication in existing NCS as well as make the system open. The application overhead is not burdensome both in event detection and traffic management. Because of Wi-Fi usage, the modification of the physical system infrastructure is also not so troublesome.

As the tests show, despite having two different communication interfaces and different communication media, there are still frequency ranges where the transmission can be disrupted. Better results can be achieved by using the frequency-hopping spread spectrum (FHSS) mechanism to reduce the radiated electromagnetic emission and increase the immunity to electromagnetic disturbances. A related possible technique is the direct sequence spread spectrum (DSSS), which also spreads the spectrum across a wide channel, but it is more susceptible to interferences and less effective as a spectrum-sharing method. Moreover, during the test, the data transmission via Wi-Fi took place through one selected channel. In the future research plan, it is intended to increase the communication immunity to EMC disturbances by applying the previously mentioned techniques FHSS, DSSS and Wi-Fi communication in the 5 GHz band.

The mutual usage of one logical network which combines two or more different physical networks allows reducing or even eliminating the communication problems caused by EMI. It is more likely that the given disruption influence reveals a negative impact on the regular

redundancy structure than on the proposed one. This research should not be considered as an alternative method of assuring safety or as something used exclusively instead of existing concepts. It is rather considered as an additional way which can be used together with redundant, mixed, and other industrial network solutions.

## Supporting information

**S1 File.**
(ZIP)

## Author Contributions

**Conceptualization:** Piotr Gaj.

**Data curation:** Michał Maćkowski.

**Formal analysis:** Piotr Gaj.

**Funding acquisition:** Piotr Gaj.

**Investigation:** Piotr Gaj, Michał Maćkowski.

**Methodology:** Piotr Gaj.

**Resources:** Michał Maćkowski.

**Software:** Piotr Gaj.

**Validation:** Michał Maćkowski.

**Writing – original draft:** Piotr Gaj, Michał Maćkowski.

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
