## [Decision Letter · Decision Letter 0]

2 Jan 2020

PONE-D-19-28357

Electromagnetic Compatibility Issues in Mixed Wired and Wireless Networked Control Systems

PLOS ONE

Dear Dr. Gaj,

Thank you for submitting your manuscript to PLOS ONE. After careful consideration, we feel that it has merit but does not fully meet PLOS ONE’s publication criteria as it currently stands. Therefore, we invite you to submit a revised version of the manuscript that addresses the points raised during the review process.

We would appreciate receiving your revised manuscript by Feb 16 2020 11:59PM. To enhance the reproducibility of your results, we recommend that if applicable you deposit your laboratory protocols in protocols.io, where a protocol can be assigned its own identifier (DOI) such that it can be cited independently in the future. For instructions see: http://journals.plos.org/plosone/s/submission-guidelines#loc-laboratory-protocols

We look forward to receiving your revised manuscript.

Kind regards,

Chi-Tsun Cheng, Ph.D., M.Sc., B.Eng.

Academic Editor

PLOS ONE

Journal Requirements:

Additional Editor Comments:

The following issues should be addressed before it can be considered for publication.

1) The comparison in section 3 and Table 1 is rather qualitative. The AE would like to see a more quantitative comparison, for example, how do those advantages be measured or calculated. What are their desirable ranges?

2) The design rationals of the experiments have not been discussed in detail. How are those parameters be chosen? How closely are they reflecting the real-life scenario?

3) Some of the evaluation criteria have not been clearly elaborated. For example in Table 2, are their BER or PER?

4) What are the implications of those numbers in Table 2? Are they significant if there are error detection and correction schemes at the higher layer of the protocol stacks?

Reviewers' comments:

Reviewer's Responses to Questions

**Comments to the Author**

1. Is the manuscript technically sound, and do the data support the conclusions?

Reviewer #1: Partly

2. Has the statistical analysis been performed appropriately and rigorously? 

Reviewer #1: No

3. Have the authors made all data underlying the findings in their manuscript fully available?

Reviewer #1: Yes

4. Is the manuscript presented in an intelligible fashion and written in standard English?

Reviewer #1: Yes

5. Review Comments to the Author

Reviewer #1: The manuscript presents a sound testing methodology and robust testing procedure. However, there are a number of concerns with the means by which results have been presented especially with regard to the different test cases chosen. These concerns are listed as follows:

- It is unclear for all test cases conducted how the "errors" generated are temporally distributed, that being respective error rates observed under each test condition. Results have only been presented indicating errors with respect to frequency specific interference that does not indicate the apparent impacts of said interference through a time-based measure (e.g. error rate).

- The results for each test often indicate that a threshold or "maximum" number of errors was reached for a particular dwell period at a certain point in each test. It is unclear how these errors or this quantity of errors impacts the application layer operation of the system.

6. PLOS authors have the option to publish the peer review history of their article (what does this mean?). If published, this will include your full peer review and any attached files.

Reviewer #1: No

---

## [Decision Letter · Decision Letter 1]

15 Apr 2020

Electromagnetic Compatibility Issues in Hybrid Wired and Wireless Industrial Networks

PONE-D-19-28357R1

Dear Dr. Gaj,

We are pleased to inform you that your manuscript has been judged scientifically suitable for publication and will be formally accepted for publication once it complies with all outstanding technical requirements.

With kind regards,

Chi-Tsun Cheng, Ph.D., M.Sc., B.Eng.

Academic Editor

PLOS ONE

Additional Editor Comments (optional):

Most reviewers' comments in the last round of review have been addressed properly. The paper is recommended to be accepted.

Reviewers' comments:

Reviewer's Responses to Questions

**Comments to the Author**

1. If the authors have adequately addressed your comments raised in a previous round of review and you feel that this manuscript is now acceptable for publication, you may indicate that here to bypass the “Comments to the Author” section, enter your conflict of interest statement in the “Confidential to Editor” section, and submit your "Accept" recommendation.

Reviewer #1: All comments have been addressed

Reviewer #2: All comments have been addressed

Reviewer #3: (No Response)

2. Is the manuscript technically sound, and do the data support the conclusions?

Reviewer #1: Yes

Reviewer #2: Yes

Reviewer #3: Partly

3. Has the statistical analysis been performed appropriately and rigorously? 

Reviewer #1: Yes

Reviewer #2: Yes

Reviewer #3: No

4. Have the authors made all data underlying the findings in their manuscript fully available?

Reviewer #1: Yes

Reviewer #2: Yes

Reviewer #3: No

5. Is the manuscript presented in an intelligible fashion and written in standard English?

Reviewer #1: Yes

Reviewer #2: Yes

Reviewer #3: No

6. Review Comments to the Author

Reviewer #1: The clarifications addressing previous concerns more clearly outlined the objectives and methodologies of the research as well as justifying their validity in relation to real time application layer constraints of a networked system.

Reviewer #2: There is no comment. All comments has been addressed.

There is no comment. All comments has been addressed.

Reviewer #3: In this paper, the authors have discussed about the mutual utilization of more than one interface. In order to back the network control system (NCS) and to manage the node related traffic within the scope of higher level service. The authors have also discussed the dependability issue from the electromagnetic compatibility point of view. The authors need to address the following points in bit more details to make the research paper more comprehensible & reliable.

Major comments:

-In case of wireless network control system, detailed discussion on interference part is missing. Please explain.

- Why mutual utilization of more than one interface is one of the major concerns? Please explain.

- To validate this type of system, measurement results of electromagnetic is needed. Otherwise, it very hard to accept that this system will work efficiently in practice.

- It would be better if authors will compare this work over existing works.

- The authors should have discussed communication issues directly related to electromagnetic interference (EMI) briefly for better understanding.

- More explanations are required in test case section. It is not clear.

- Big picture is missing

- Experimental setup is missing.

Minor comments:

• The explanation of figure (5) is not adequate. Elaborate the significance of (0,70) point of in the given graph.

• In sub-subsection 4.3.3, the last line of the paragraph under the point (d) is not properly aligned.

• Overall presentation must be improved.

7. PLOS authors have the option to publish the peer review history of their article (what does this mean?). If published, this will include your full peer review and any attached files.

Reviewer #1: No

Reviewer #2: No

Reviewer #3: No

---

## [Editor Report · Acceptance letter]

22 Apr 2020

PONE-D-19-28357R1 

Electromagnetic Compatibility Issues in Hybrid Wired and Wireless Industrial Networks 

Dear Dr. Gaj:

I am pleased to inform you that your manuscript has been deemed suitable for publication in PLOS ONE. Congratulations! Your manuscript is now with our production department. 

With kind regards,

on behalf of

Dr. Chi-Tsun Cheng 

Academic Editor

PLOS ONE